# Prognostic Role of Circulating Tumor Cells in Patients with Metastatic Castration-Resistant Prostate Cancer Receiving Cabazitaxel: A Prospective Biomarker Study

**DOI:** 10.3390/cancers15184511

**Published:** 2023-09-11

**Authors:** Filippos Koinis, Zafeiris Zafeiriou, Ippokratis Messaritakis, Panagiotis Katsaounis, Anna Koumarianou, Emmanouil Kontopodis, Evangelia Chantzara, Chrissovalantis Aidarinis, Alexandros Lazarou, George Christodoulopoulos, Christos Emmanouilides, Dora Hatzidaki, Galatea Kallergi, Vassilis Georgoulias, Athanasios Kotsakis

**Affiliations:** 1Department of Medical Oncology, University General Hospital of Larissa, 41334 Larisa, Greece; valiaxantzara@gmail.com (E.C.); valadisaidarinis@yahoo.com (C.A.); lazaroualexander@gmail.com (A.L.); gchristodoulopoulos@hotmail.gr (G.C.); 2Faculty οf Medicine, School of Health Sciences, University of Thessaly, 41335 Larissa, Greece; 3Second Department of Medical Oncology, Theageneion Anticancer Hospital, 54007 Thessaloniki, Greece; zafeiris.zafeiriou@outlook.com; 4Laboratory of Tumor Cell Biology, School of Medicine, University of Crete, Heraklion, 70013 Crete, Greece; i_messaritakis@yahoo.com; 5First Department of Medical Oncology, Metropolitan General Hospital, 15562 Athens, Greece; pvkatsaounis@gmail.com (P.K.); georgulv@otenet.gr (V.G.); 6Medical Oncology Unit, 4th Department of Internal Medicine, “ATTIKON” University Hospital of Athens, 11528 Athens, Greece; akoumari@yahoo.com; 7Department of Medical Oncology, “Venizelion” General Hospital of Heraklion, 71409 Crete, Greece; kontopodise@gmail.com; 8Department of Medical Oncology, Diavalkanikon General Hospital of Thessaloniki, 55535 Thessaloniki, Greece; chrem@interbalkan-hosp.gr; 9Hellenic Oncology Research Group (HORG), 11526 Athens, Greece; dhatzidaki@gmail.com; 10Division of Genetics, Cell and Developmental Biology, Department of Biology, University of Patras, 26504 Patras, Greece; gkallergi@upatras.gr

**Keywords:** cabazitaxel, CTCs, prognostic, prostate cancer, liquid biopsy, CRPC

## Abstract

**Simple Summary:**

This prospective translational study evaluated whether the presence of circulating tumor cells (CTCs) in the peripheral blood (PB) correlates with efficacy of cabazitaxel in patients with metastatic castration-resistant prostate cancer (mCRPC). This study demonstrated that 84% of patients with mCRPC have at least one CTC/7.5 mL in their PB and that those patients with at least 5 CTCs/7.5 mL of blood have an increased risk of early death. In mCRPC patients under treatment with cabazitaxel, CTC counts both at baseline and after the first cycle display prognostic significance. Studies incorporating CTC counts in the prognostic and predictive algorithms of mCRPC are warranted.

**Abstract:**

Rational: Circulating tumor cells (CTCs) appear to be a promising tool for predicting the clinical outcome and monitoring the response to treatment in patients with solid tumors. The current study assessed the clinical relevance of monitoring CTCs in patients with metastatic castration resistant prostate cancer (mCRPC) treated with cabazitaxel. Patients and Methods: Patients with histologically confirmed mCRPC who were previously treated with a docetaxel-containing regimen and experienced disease progression were enrolled in this multicenter prospective study. CTC counts were enumerated using the CellSearch system at baseline (before cabazitaxel initiation), after one cabazitaxel cycle (post 1st cycle) and at disease progression (PD). Patients were stratified into predetermined CTC-positive and CTC-negative groups. The phenotypic characterization was performed using double immunofluorescence staining with anti-CKs and anti-Ki67, anti-M30 or anti-vimentin antibodies. Results: The median PFS and OS were 4.0 (range, 1.0–17.9) and 14.5 (range, 1.2–33.9) months, respectively. At baseline, 48 out of 57 (84.2%) patients had ≥1 CTCs/7.5 mL of peripheral blood (PB) and 37 (64.9%) had ≥5 CTCs/7.5 mL of PB. After one treatment cycle, 30 (75%) out of the 40 patients with available measurements had ≥1 detectable CTC/7.5 mL of PB and 24 (60%) ≥ 5CTCs/7.5 mL of PB; 12.5% of the patients with detectable CTCs at the baseline sample had no detectable CTCs after one treatment cycle. The detection of ≥5CTCs/7.5 mL of PB at baseline and post-cycle 1 was associated with shorter PFS and OS (*p* = 0.002), whereas a positive CTC status post-cycle 1 strongly correlated with poorer OS irrespective of the CTC cut-off used. Multivariate analysis revealed that the detection of non-apoptotic (CK^+^/M30^−^) CTCs at baseline is an independent predictor of shorter OS (*p* = 0.005). Conclusions: In patients with mCRPC treated with cabazitaxel, CTC counts both at baseline and after the first cycle retain their prognostic significance, implying that liquid biopsy monitoring might serve as a valuable tool for predicting treatment efficacy and survival outcomes.

## 1. Introduction

Prostate cancer (PCa) remains the most common malignancy diagnosed and the second most common cause of cancer death among men in developed countries, with approximately 288,300 expected new cases predicted to count for 11% of all cancer deaths in the US during 2023 [1]. The life-prolonging, widely used, and approved therapies for PCa are based on targeting androgen signaling, bone targeting with radiopharmaceuticals, and chemotherapy. Among them, cabazitaxel, a tubulin-binding taxane, had shown robust antitumor activity in preclinical models resistant to paclitaxel and docetaxel in the early 2000s [2]. Based on these encouraging preliminary results, a randomized phase III trial (TROPIC trial) in men with mCRPC progressing during or after treatment with docetaxel was conducted. Patients were randomly allocated to mitoxantrone plus prednisone or cabazitaxel (25 mg/m^2^) plus prednisone. This trial demonstrated a significant prolongation of overall survival (OS) in favor of cabazitaxel, with a hazard ratio (HR) of 0.7 for death [median OS (mOS): 15.1 versus 12.7 months in the cabazitaxel and mitoxantrone groups, respectively; *p* < 0.0001]. Median progression-free survival (mPFS) was also improved (2.8 versus 1.4 months in the cabazitaxel and mitoxantrone groups, respectively; HR 0.74, 0.64–0.86, *p* < 0.0001). The most common clinically significant grade 3 or higher adverse events were neutropenia and diarrhea [3]. In 2017, results from the PROSELICA trial suggested that a lower-dose regimen (20 mg/m^2^) was noninferior for OS and was associated with a more favorable safety profile compared to the TROPIC trial regimen [4]. These data led to the FDA and EMA granting the approval of cabazitaxel as salvage treatment of patients with mCRPC after first-line treatment with docetaxel. Nonetheless, the targets in PCa management have not changed over the years, and current treatment options have already reached their potential. Refinement in existing strategies will likely result in incremental improvement in survival rates, but major advances in therapy will likely require novel treatments approaches based on alternative biological mechanisms. In line with this are the encouraging observations reported with ^177^Lu-PSMA-617 [5] and immune checkpoint blockade in select patient subsets [6], as well as with the use of PARP inhibitors in cancers with DNA damage repair gene alterations [7]. However, despite the marked lengthening of OS, cure rates in men with metastatic PCa remain low. This result is not surprising, given the absence of biomarkers to guide the selection of patients and monitor treatment efficacy. Considering the biological heterogeneity of mCRPC and the fact that PSA is more a tissue-specific and not a cancer-specific molecule, PSA testing is far from being considered the ideal biomarker and has not been shown to be a surrogate of survival at an individual-patient level [8]. Thus, an emerging critical unmet need is to identify novel candidate markers for survival and response to treatment in vitro, validate their prognostic and predictive significance in vivo and leverage the understanding for informing the application and monitoring the efficacy of treatment clinically.

PCa biomarkers can be isolated either from prostate tissue or from serum, urine, and prostatic fluid. Recently, liquid biopsies have emerged as a convenient, minimally invasive option for obtaining tumor-derived biological information. Advances in this field of molecular interrogation of liquid samples have expanded our research capabilities for monitoring tumor growth, enabling a personalized oncology approach [9]. Converging clinical and experimental observations suggest that CTCs could serve as a liquid biopsy approach allowing the collection of molecular and genetic information to provide a real-time, updated, and more relevant assessment of tumor status than the analysis of tissue sample obtained during the initial diagnosis. Since their identification in 1869 by Ashworth, CTCs’ phenotypic characterization and enumeration have enthusiastically been embraced and widely studied by the scientific community. Nowadays, the potential prognostic and predictive role of CTCs has been appreciated across various cancer types. CTCs have been shown to act as a driving force of cancer initiation and progression, contributing to resistance to treatment [10]. Moreover, a clear-cut correlation between the absolute number of CTCs in the peripheral blood and the clinical outcome of patients with various tumor types has also been confirmed [11,12,13]. These findings account for the interest in the potential impact of CTCs on the clinical outcome of patients with PCa.

In search of the optimal setting for studying CTCs, their clinical utility in patients with localized/early PCa has been hampered due to their scarcity. In contrast, serial CTCs analysis through the CellSearch assay has received FDA approval for use in conjunction with other clinical methods (imaging, etc.) for monitoring patients with metastatic PCa [14]. Notably, in patients with advanced disease, the presence of CTCs in the peripheral blood before treatment initiation is associated with shorter PFS and OS. Specifically, in a pivotal prospective study patients with mCRPC and ≥5 CTCs/7.5 mL of PB prior to the initiation of chemotherapy had significantly shorter OS compared to those with <5 CTCs (10 versus 21 months, respectively). Moreover, post-treatment CTCs counts were shown to better correlate with OS than PSA decrements [15]. In a pre-specified analysis of a phase III trial of abiraterone plus prednisone compared to placebo plus prednisone in patients with mCRPC progressing after docetaxel treatment, patients with <5 CTCs/7.5 mL had significantly longer OS than those with ≥5 CTCs/7.5 mL, in both treatment arms. Moreover, CTC conversion (defined as a decline of CTC counts from ≥5 to <5/7.5 mL) emerged as a key independent prognosticator of OS [16]. 

The aim of the current correlative translational research study was to evaluate the effect of second line cabazitaxel on the changes in CTCs status in patients with mCRPC, determine a potential association between those changes and the patients’ clinical outcome, and assess the clinical relevance of specific CTCs’ subpopulations in this setting.

## 2. Materials and Methods

### 2.1. Study Design

Patients with metastatic, histologically, or cytologically confirmed adenocarcinoma of the prostate that was progressing despite castrate levels of testosterone (<50 ng/mL) and who were previously treated with a docetaxel-containing regimen were enrolled in the study. Additional eligibility criteria included age ≥18 years, adequate liver, renal, and bone marrow function, and an ECOG performance status of ≤2. At least a four-week washout period since the last administration of docetaxel was required. Abiraterone acetate or enzalutamide, either pre or post docetaxel first-line therapy, were allowed. In case of treatment with abiraterone acetate or enzalutamide after docetaxel failure, the drug must have been discontinued for at least four weeks before enrollment in the study. CNS metastases were allowed if they had been irradiated and the patient was clinically stable. Patients with a history of other malignancy within the last 5 years, a history of radiotherapy/surgery within 4 weeks prior to enrollment date, or a history of severe hypersensitivity reaction to docetaxel or to polysorbate 80-containing drugs were excluded. Patients were treated in the Hellenic Oncology Research Group’s (HORG) collaborative centers. The study was approved by the Scientific and Ethics Committees of the participating Institutions [University Hospital of Heraklion, Crete, Greece (5729-30/4/2013), General Maternal Hospital of Athens “Elena Venizelou”, Athens, Greece (375-8/6/2015) and 251 Air Force General Hospital of Athens, Athens, Greece (4508-4/12/2014) and University General Hospital of Larissa, Larissa, Greece (2003-9/4/2020)] and all experiments were performed in accordance with relevant guidelines and regulations. All patients provided written informed consent to participate in the study and had the right to withdraw consent at any time.

### 2.2. Study Procedures

Ten collaborative Greek centers enrolled patients in the study. Patients received oral prednisone 10 mg daily and cabazitaxel 25 mg/m² intravenously over 1 h on day 1 of each 21-day cycle according to the standard clinical practice and national guidelines and were followed prospectively. Treatment was administered until PD, occurrence of unacceptable toxicity, or consent withdrawal. Pretreatment evaluations included a detailed medical history, physical examination, laboratory screening (complete blood count and biochemistry profile including lactate dehydrogenase and alkaline phosphatase), serum PSA concentration, bone scan, and computed tomography (CT) as clinically indicated. Physical examination and blood tests (including PSA) were repeated before each infusion of cabazitaxel and at the end of treatment. Imaging studies (bone scan, CT) were performed every 3 treatment cycles. 

### 2.3. Blood Samples and CTC Isolation

For the evaluation of CTCs, PB was obtained at 3 prespecified timepoints: (i) before the administration of the first cabazitaxel cycle and at least 3–4 weeks from the last administration of biphosphonates or denosumab (baseline sample), (ii) before the administration of the second chemotherapy cycle (post 1st sample), and (iii) at the time of PD. PB was obtained into both 10 mL preservative tubes for CTC enumeration (Menarini Silicon Biosystems, USA) and 20 mL EDTA-containing tubes for CTC phenotypic characterization. All blood samples were obtained at the middle-of-vein puncture after the first 5 mL of blood were discarded to avoid contamination with epithelial cells from the skin. Samples were maintained at room temperature and processed within 24 h of collection. CTC enumeration was performed using the CellSearch system. The common definition of CTCs was applied as 4’,6-diamidino-2-phenylindole (DAPI)-positive nucleated cells, negative for CD45, and cytokeratin (CK)-positive, obtained through positive selection based on epithelial cell adhesion molecule (EPCAM) expression [15]. 

For phenotypic characterization, peripheral blood mononuclear cells (PBMCs) were isolated by Ficoll–Hypaque density (d = 1077 g/mL; Sigma-Aldrich Chemie GmbH, Taufkirchen, Germany) centrifugation at 1800 rpm for 30 min at ambient temperature. Aliquots of 5 × 10^5^ PBMCs were cyto-centrifuged at 2000 rpm for 2 min on glass microscope slides. Cytospins were dried and stored at −80 °C until use. Two slides (10^6^ PBMCs) from each patient were analyzed at each time point. The presence of CTCs in cytospins of PBMCs was evaluated using the appropriate antibodies: monoclonal antibodies against Ki67 (a proliferation marker; Abcam, Cambridge, UK), M30 (an apoptosis marker; CytoDEATH fluorescein, Roche, Manheim, Germany), and vimentin (an EMT marker; Santa Cruz, Santa Cruz, CA, USA), as previously described [17]. The mouse A45-B/B3 antibody (detecting CK8, CK18 and CK19 and the stained cells will be referred as CK^+^ in the text; Micromet, Munich, Germany) was used to confirm the epithelial origin of the cells. The cytomorphological criteria described by Meng et al. [18] were used for the characterization of a CK^+^ cell as a CTC.

### 2.4. Double Immunofluorescence Assay

Double immunofluorescence staining was performed as described previously [19]. Briefly, PBMCs cytospins were fixed with 3% paraformaldehyde and incubated with the appropriate primary and secondary antibodies for one hour. Vimentin and Ki67 were labelled with anti-rabbit Alexa 555 (Molecular Probes, Invitrogen, Carlsbad, CA, USA), M30 was an fluorescein-conjugated mouse antibody, and CK was detected using the corresponding secondary fluorescein isothiocyanate (FITC) fluorochrome or anti-mouse Alexa 555 (Molecular Probes). Finally, samples were mounted on DAPI antifade reagent (Molecular Probes) for cell nuclei visualization. Negative controls were prepared by omitting one of the first antibodies (anti-Ki67, anti-M30, anti-Vim, anti-CK), but incubating the cells with the respective secondary antibody. Slides were analyzed using a fluorescence microscope (Leica DM 2500, Heidelberg, Germany) by two experienced biologists and the results are expressed as number of CTCs/10^6^ PBMCs.

### 2.5. Statistical Considerations

This was a prospective translational study designed to assess the clinical relevance of CTC changes during the administration of cabazitaxel plus prednisone as salvage treatment in pre-treated mCRPC patients as per its labeled indication, as well as the efficacy and safety of the treatment in the real-life practice setting. We hypothesized that the PSA response rate of the patient population under study after treatment would be expected to be around 40%. Then, at a significance level α = 0.05, the required sample size for achieving an 80% power correctly detecting a difference of 22% would mean a total number of 59 patients had to be enrolled in the study. PSA response was defined as ≥50% decline from baseline (measured twice 3 to 4 weeks apart) with no evidence of disease progression on imaging. Overall response rate (ORR) was defined as the proportion of patients with complete or partial response observed during the first 18 weeks of study treatment according to RECIST version 1.1. PFS was defined as the time elapsed between the start date of treatment and the date of clinical or radiological progression or death from any reason. OS was defined as the time elapsed between the start date of treatment until the date of death from any reason or the date of last follow-up. The evaluation for the presence of positive cells was performed in a blinded manner to clinical data. PFS and OS for all patients were estimated using the Kaplan–Meier analysis and the comparisons were computed with the log-rank test. Univariate and multivariate Cox proportional hazards regression models with hazard ratios (HR) and 95% CIs were used to assess the association between potential prognostic factors and PFS or OS. All statistical tests were two-sided and *p*-values < 0.05 were considered statistically significant. Data were analyzed using the SPSS statistical software, version 22.0 (SPSS Inc., Chicago, IL, USA).

Qualitative factors were compared by Pearson’s Chi-square test or Fisher’s exact test, whenever appropriate. Differences in positivity rates were assessed using the McNemar test and differences in terms of continuous variables were assessed by the non-parametric Wilcoxon test.

## 3. Results

### 3.1. Patient Characteristics

A total of 60 patients with progressing mCRPC were screened. Fifty-nine were enrolled, since one screened patient withdrew his consent. The patients’ clinical characteristics are summarized in Table 1. The median age was 67 years (range 44–84), and 48 (81.3%) had an ECOG PS of 0–1. Of the 52 patients with known Gleason score, all but one patient had a Gleason score of >6 (7–10). All patients had previously received a novel androgen receptor targeting agent (ARTA) and taxane-based chemotherapy. Notably, fourteen (23.7%) patients had visceral metastasis and the median number of involved metastatic sites was two (range, 1–5). Cabazitaxel was administered as second-line treatment for mCRPC in 14 (23.7%) patients and greater than second-line in the remaining patients. The majority (69.5%) of the patients had high baseline serum PSA levels (>20 ng/mL).

### 3.2. Drug Exposure and Efficacy 

A total of 345 chemotherapy cycles were administered for a median number of 5 (range, 1–16) cycles. Dose reduction was required in three patients because of hematologic (n = 2) and non-hematologic (n = 1) adverse events. At the time of analysis, 32 (54.2%) of the 59 evaluable patients had died and 58 (98.3%) patients had discontinued treatment because of disease progression (n = 50), adverse events (n = 1), toxicity-related death (n = 1), and other reasons (i.e., physicians’ choice; n = 6). After a median follow-up period of 17.5 months (range, 1.2–33.9), the median PFS for the treated patients was 4.0 [range, 1.0–17.9 months; 95% confidence interval (95% CI), 3.5–4.5 months] months while the median OS was 14.5 (range, 1.2-33.9 months; 95% CI, 8.9–20.2 months) months. The 1-year survival rate was 55.2%.

### 3.3. CTC Status at Baseline (Pre-Cabazitaxel Sample)

At baseline, 57 patients had adequate biological material for the evaluation of CTC status; 2 (3.4%) patients were not evaluable because no CTC enumeration could be performed due to technical problems. In total, 48 (84.2%) patients had ≥1 CTCs/7.5 mL of PB, whereas 15.8% had no CTCs. However, by adopting a cut-off of 5 CTCs/7.5 mL of PB as a threshold to dichotomize the patient population, 37 (64.9%) of them were considered as CTC-positive (Table 2). The median number of detectable CTCs at baseline was 13/7.5 mL of PB (range, 0–3107).

### 3.4. CTC Status after 1st Cabazitaxel Cycle (Post 1st Cycle Sample)

Forty patients (67.8%) had a second blood sample 21 days after the administration of the first chemotherapy cycle. Patients that had adequate biological material both at baseline and after the 1st chemotherapy cycle were included in this analysis. As shown in Table 2, 30 patients had at least 1 detectable CTC/7.5 mL of PB, whereas 24 had ≥5 detectable CTCs/7.5 mL of PB; the median number of detectable CTCs at this timepoint was 8.5 CTCs/7.5 mL (range, 0–1733). There was not a statistically significant decrease in the mean number of CTCs at the post-1st cycle time point compared to the baseline sample (*p* = 0.963; Wilcoxon test).

Five (12.5%) patients with detectable CTCs (≥1 CTCs/7.5 mL of PB) in the baseline sample turned CTC-negative in the post-1st cycle sample. On the contrary, 2 (5%) patients with 1–4 detectable CTCs/7.5 mL of PB in the baseline sample had increased the number of detectable CTCs of ≥5CTCs/7.5 mL of PB in the post-1st sample. In total, 5 patients (5/27 patients; 18.5%) had a reduction from ≥ 5 to ≤ 4 CTCs/7.5 mL of PB (Figure 1).

### 3.5. CTC Status at Disease Progression (PD Sample)

Similarly, the CTC status was evaluated in 20 patients at the time of disease progression; among them, 17 (85%) and 13 (65%) had ≥1 and ≥5 CTCs/7.5 mL of PB (range, 0–228) (Table 2). The median number of CTCs at PD was 9.5 CTCs/7.5 mL of PB. There was no statistically significant change in the mean number of CTCs between the PD samples and the baseline or the post 1st cycle samples.

### 3.6. CTC Status and Clinical Outcomes

All patients included in the CTC analysis were evaluated for response assessment. No statistically significant difference was found in terms of the response rate (RR) between patients below or above the cut-off values of ≥1 or ≥5 CTCs/7.5 mL of PB. However, patients without detectable CTCs at baseline had a higher probability to experience disease control (PR and SD) compared to patients with detectable [≥1 CTCs/7.5 mL of PB (*p* = 0.099)], (Appendix A). Nonetheless, a difference in terms of RR according to the detection or not of CTCs after one chemotherapy cycle could not be observed, irrespective of the used cut-off. Similarly, there was no statistically significant difference in terms of PFS between the CTC-defined patients’ subgroups, regardless of the threshold used for CTC positivity. Nevertheless, we observed a non-significant trend, since patients with ≥5 CTCs/7.5 mL of PB at baseline had a higher probability for disease progression compared to patients with <5 CTCs/7.5 mL of PB (*p* = 0.089), as CTC-positive patients had a shorter PFS (3.7 months; 95% CI, 3–4.4 months) compared with CTC-negative patients (5.4 months; 95% CI, 4.5–6.2 months), (Appendix A). Notably, the detection of CTCs both at baseline and after the 1st cycle of cabazitaxel was associated with a significantly shorter OS; in particular, using the cut-off of 5 CTCs/7.5 mL of PB, CTC-negative patients at baseline had significantly higher OS (28 months; 95% CI, 11.4–44.6 months) compared to CTC-negative patients (8.8 months; 95% CI, 3.1–14.5 months) (*p* = 0.002; Table 3; Figure 2). This difference was not observed when the threshold of 1 CTCs/7.5 mL of PB was used. Interestingly, a positive CTC status after the administration of one chemotherapy cycle strongly correlated with poorer OS irrespective of the cut-off used (*p* = 0.047, *p* = 0.005) (Table 4; Figure 3a,b). Due to the small number of patients, CTC status at disease progression was not associated with clinical outcomes in this study (Appendix A).

### 3.7. Phenotypic Characterization of CTCs 

Additional exploratory analyses aimed to determine the incidence of different CTC subpopulations before the initiation (baseline sample) and after one cabazitaxel cycle based on the expression of Ki67, M30, and vimentin (Table 5). Indeed, proliferating CTCs (CK^+^/Ki67^+^) could be detected in 85.7% and in 71.4% of patients at baseline and after one chemotherapy cycle, respectively. Moreover, 80% and 85.7% of patients harbored non-apoptotic CTCs (CK^+^/M30^−^) at baseline and after one chemotherapy cycle, respectively. Similarly, 94.3% and 95.2% of the patients had Vim^+^ at baseline and after one treatment cycle, respectively. Univariate analysis demonstrated that the presence of non-apoptotic (CK^+^/M30^−^) CTCs was associated with a significantly shorter PFS (*p* = 0.005) and OS (*p* = 0.003) compared to patients with apoptotic (CK^+^/M30^+^) CTCs (Table 6). In addition, multivariate analysis revealed that the detection of non-apoptotic (CK^+^/M30^−^) CTCs is an independent factor associated with shorter OS (*p* = 0.005; Table 6). There was no association between the detection of CK^+^/Ki67^+^ and CK^+^/Vim^+^ and PFS or OS (Table 6).

## 4. Discussion

Several avenues of research investigating potential clinical applications of CTCs characterization and enumeration have provided sufficient data that led to the FDA approval of CellSearch^®^ detected CTCs for prognostication of patients with advanced PCa. The present study adds to the growing body of evidence that CTC counts are prognostic and predict OS in mCRPC. This multicenter prospective study specifically showed that CTC number at different time points after treatment was the strongest independent predictor of OS in mCRPC. These data underline the prognostic significance of baseline CTCs and for the first time, to our knowledge, demonstrate that the sheer number of CTCs predicts survival after the first cycle of treatment with cabazitaxel. In addition, although CTCs heterogeneity has been previously described in patients with mCRPC, our study is among the few to describe the association between the detection of different CTC phenotypes (apoptotic and proliferating) and the clinical outcome.

Previous studies have reported that the detection of CTCs at baseline is associated with both DFS and OS in patients with CRPC. In a landmark study, using the cut-off values of ≥5 CTC/7.5 mL of PB versus <5 CTC/7.5 mL of PB to define unfavorable and favorable groups, de Bono et al. reported that the number of CTCs in the blood of patients with mCRPC before treatment initiation is prognostic [15]. Notably, the IMMC38 trial led to the FDA approval of CTC enumeration using the CellSearch platform in patients with CRPC. Similarly, the phase III MAINSAIL trial revealed that the detection of ≥5 CTCs/7.5 mL of PB at baseline in mCRPC patients treated with docetaxel was associated with poor OS, but did not correlate with tumor (RECIST criteria) or PSA response [20]. In agreement with the aforementioned data, preplanned analyses of the AFFIRM [21] and COU-301 [16] trials further supported the favorable clinical outcome of patients with less than 5 CTCs/7.5 m: of PB. To thoroughly examine the underlying biology of CTCs, the prospective PROPHECY trial addressed the clinical significance of CTCs expressing the androgen receptor splice variant AR-V7, in the PB of patients with mCRPC. A cut-off of 5 CTCs/7.5 mL of PB was used to stratify patients into CTC-positive and CTC-negative subgroups. Men with AR-V7 positive status at baseline exhibited worse PFS and OS, if treated with enzalutamide or abiraterone, but could still derive benefit from taxane chemotherapy [22]. Since then, other classifications (>1, >3, >50 CTCs) or the absolute number of CTCs at baseline have also been examined for their prognostic value. Danila et al. reported that the number of CTCs at baseline correlated strongly with survival, but no threshold effect could be identified [23], suggesting that CTC counts are important as a continuous variable in predicting the clinical outcome of patients with mCRPC. To this end, Olmos et al., demonstrated that a cut-off value of 50 CTCs/7.5 mL of PB is a predictor of poorer survival than that of 5 CTCs/7.5 mL [24]. Finally, de Kruijff et al. have assessed baseline CTC numbers in 114 patients with mCRPC receiving cabazitaxel therapy. Using a cut-off of ≥5 CTCs/7.5 mL of PB, they reported a strong correlation between CTC counts and survival outcomes [25]. In agreement with these data, our results in pre-treated patients with mCRPC demonstrate a clear-cut association between the detection of ≥5 CTCs/7.5 mL of PB at the baseline sample and shorter mOS. This was not observed when the cut-off ≥1 CTCs/7.5 mL of PB was used. Thus, our cut-off analysis did confirm the previously reported negative prognostic role of more than 5 CTCs/7.5 mL of PB for patients with mCRPC treated with cabazitaxel.

Despite showing that the pre-treatment absolute number of CTCs in the PB of a patient with mCRPC is a simple, minimally invasive method for assessing the clinical outcome, it was yet to be determined whether CTC detection after one cycle of treatment constitutes an early predictor of response to cabazitaxel. Moreover, to date, there is no information regarding the effect of one cycle of cabazitaxel on CTCs in patients with mCRPC. The presented data clearly indicate that the detection of CTCs after the administration of one chemotherapy cycle was associated with a significantly decreased OS, irrespective of the CTC cut-off use. Thus, post-1st cycle CTC counts predict survival after treatment. Conversely, there was not any correlation with the detection of CTCs and the PFS or RR. Although the mPFS of patients with ≥5 CTCs/7.5 mL of PB after one chemotherapy cycle was shorter than the mPFS of patients with <5 CTCs/7.5 mL of PB, the difference was not statistically significant, probably due to the limited cohort sizes. Nevertheless, a tendency for CTCs to decrease under systemic therapy was evident. As shown in Figure 1, only two patients experienced an increase in the number of detected CTCs and changed from the <5 CTCs/7.5 mL of PB to the ≥5 CTCs/7.5 mL. Intriguingly, both patients had disease progression at the first radiological evaluation of the treatment outcome. The increase of CTC counts after one cabazitaxel cycle predicted for early tumor progression, but the small number of patients precludes any definitive conclusions. Nevertheless, we found a reduction in the median number of detected CTCs after one treatment cycle, indicating that cabazitaxel had a limited effect on the CTC populations in the PB of patients with mCRPC. This observation strongly suggests that cabazitaxel can eliminate a clinically significant proportion of CTCs, but whether this effect is directed against a specific CTC subpopulation is unclear at the present time. Indeed, the frequency of the detection of apoptotic and non-apoptotic CTCs or proliferating and non-proliferating CTCs remained practically unchanged after one treatment cycle. These findings support the hypothesis for the CTC-targeting effect of cabazitaxel and reinforce the clinical utility of this drug in treating patients with mCRPC. In addition, monitoring CTC levels during treatment with cabazitaxel could reveal a proportion of patients with refractory disease who may benefit from an early CTC-informed treatment discontinuation. This could be achieved both by allowing patients to receive a potentially more effective drug (e.g., 177Lu-PSMA-617 for patients with PSMA positron emission tomography (PET) scan-positive mCRPC, pembrolizumab for patients with MSI-high tumors, metronomic cyclophosphamide, platinum-based chemotherapy, docetaxel rechallenge, or inclusion in clinical trial) and by minimizing the toxicity from ineffective treatment. Whether such a strategy could lead to an OS benefit remains to be answered by large randomized prospective clinical trials.

In addition to CTC enumeration, deeper molecular interrogation to further appreciate the phenotypic CTC heterogeneity offers another promising tool for prognostication. To date, most studies have focused on tumor cell heterogeneity and the impact of detecting specific markers on clinical behavior, therapy resistance and survival of patients with PCa [26,27]. However, in the past decade, emerging technologies for CTC isolation and characterization have permitted research on the biology of CTCs. These studies have identified distinct molecular subgroups of CTCs in patients with advanced PCa providing insights into inter-patient heterogeneity and subsequently potential targets for treatment [28,29]. The detection of epithelial–mesenchymal transition (EMT) markers on CTCs has been increasingly demonstrated to correlate with poor survival and resistance to treatment [30]. Based on ample evidence for an association between Ki67 and vimentin expression at the tumor site and poor clinical outcome of patients with mCRPC, Lindsay et al. studied the expression of these molecules on CTCs. Patients with either vimentin^+^ or Ki67^+^ CTCs at baseline had significantly poorer survival compared to those without. However, no significant changes were reported in the counts of Ki67^+^ and vimentin^+^ CTCs between the pre- and post-treatment samples [31]. Similarly, in our study, cabazitaxel had no effect on the different CTC subpopulations. Indeed, the frequency of the detection of the different CTC subpopulations (Ki-67-positive versus negative, vimentin-positive versus negative and M30-positive versus negative) remained practically unchanged after one treatment cycle. Nevertheless, probably due to the limited cohort sizes, our data did not confirm the prognostic role of EMT (vimentin^+^) and proliferating (Ki67^+^) CTCs. However, the detection of non-apoptotic CK^+^/M30^-^ cells was associated with the clinical outcome, both in terms of PFS and OS. Thus far, limited data have been available regarding the apoptotic and viable component of CTCs in patients with mCRPC. Although the detection of M30^+^ CTCs has been proven to be feasible [32], there is a paucity of information regarding their prognostic role. While increased numbers of M30^+^ CTCs have been associated with poor prognosis in patients with metastatic breast cancer [33], to our knowledge, this is first study demonstrating the prognostic significance of this CTC subgroup in patients with mCRPC. These observations raise the possibility that studying molecularly defined CTC subpopulations may form a biologically informed approach for a more comprehensive monitoring of tumor dynamics.

Although encouraging, our data must be interpreted as hypothesis-generating, as this study included a limited number of patients. Even though we observed several significant correlations, some did not reach the level of statistical significance due to the small sample size. Moreover, our ability to collect serial blood samples for harvesting CTCs at the different timepoints was hampered by patients being lost to follow-up and patients with rapidly deteriorating general condition. These limitations further added to the inherent shortcomings of the CellSearch system [34], making comparisons between patient subgroups difficult to interpret. Larger prospective studies incorporating CTC subpopulations counts in the prognostic and the predictive algorithms of mCRPC are warranted. Moreover, whether integration of ctDNA monitoring could be of complementary prognostic value remains to be elucidated.

## 5. Conclusions

In conclusion, as a form of liquid biopsy, monitoring CTCs during treatment with cabazitaxel in patients with mCRPC holds great value in predicting clinical outcomes. Future research focused on the clinical significance of the various CTC subpopulations is of paramount importance in facilitating the implementation of precision medicine in mCPRC.

## Figures and Tables

**Figure 1 cancers-15-04511-f001:**
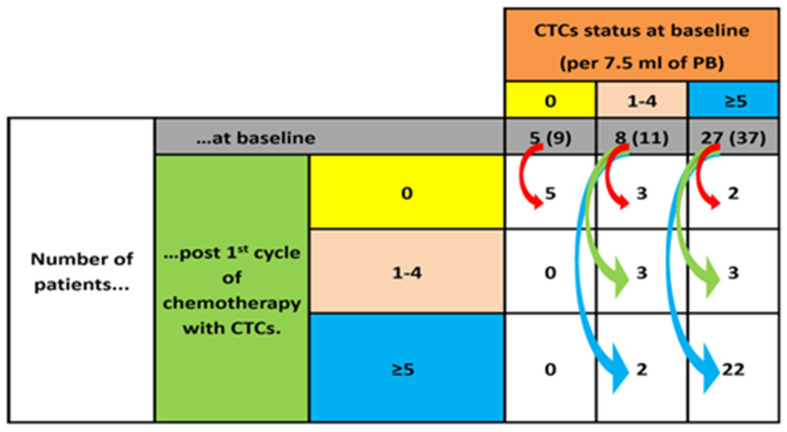
Changes in CTC counts after the first cycle of chemotherapy. For each CTC count category at baseline (0, 1–4 and ≥5 CTCs; horizontal colored boxes), the number of patients belonging to each category is shown (grey boxes). The arrows show how these patients migrate to different CTC categories (vertical-colored boxes) after the first cycle of chemotherapy. Patients that had adequate biological material both at baseline and after the first chemotherapy cycle were included in this analysis. The number in the brackets refers to the number of patients that were analyzed for their baseline sample.

**Figure 2 cancers-15-04511-f002:**
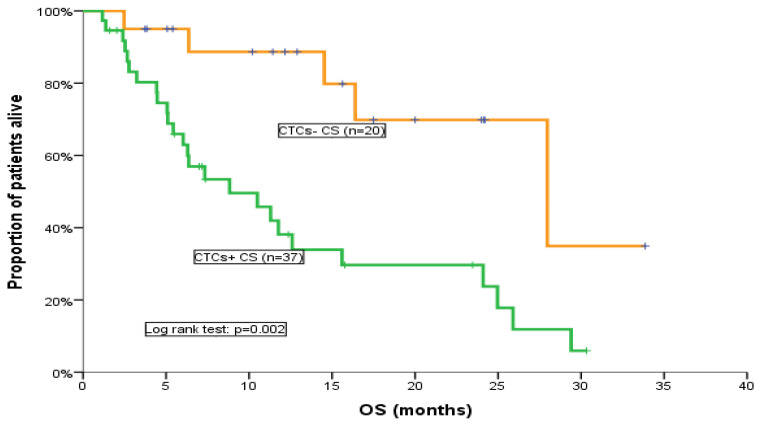
OS for CTC-positive and CTC-negative patients at baseline (cut-off: 5 CTCs/7.5 mL of PB).

**Figure 3 cancers-15-04511-f003:**
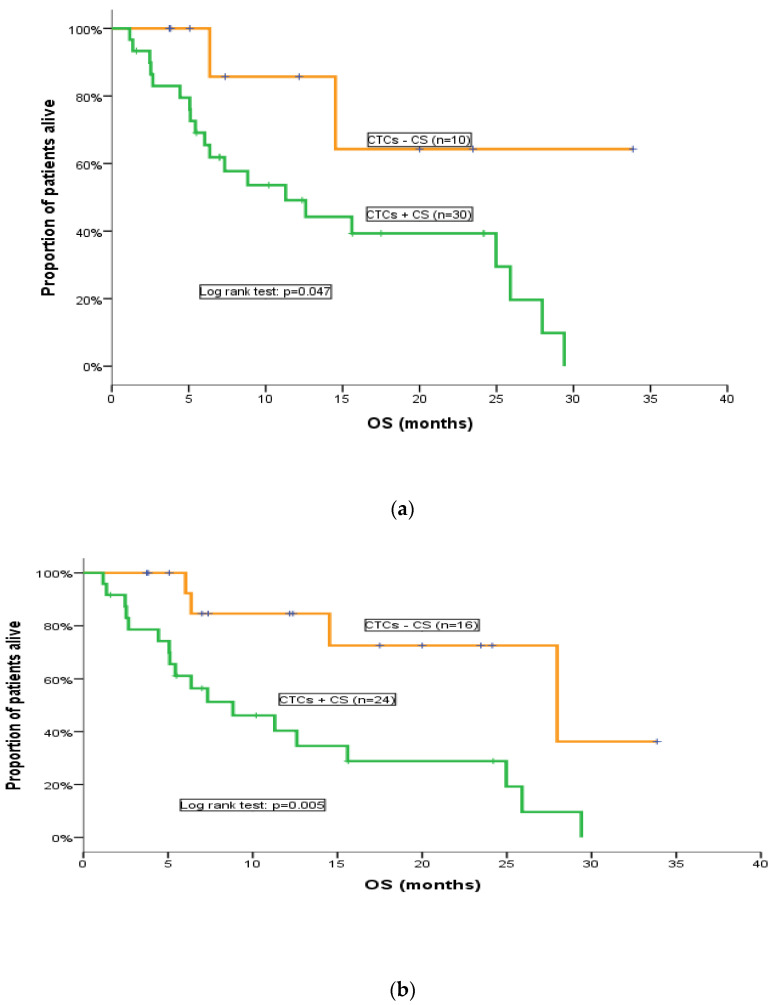
OS for CTC-positive and CTC-negative patients after the first chemotherapy cycle. (**a**) (cut-off: 1 CTCs/7.5 mL of PB); (**b**) (cut-off: 5 CTCs/7.5 mL of PB).

**Table 1 cancers-15-04511-t001:** Demographic data.

	N (59)	%
Age at diagnosis Median (min-max)	67.0 (44–84)	
Age at study enrollment Median (min-max)	75.0 (54–86)	
Performance status		
0	14	23.7
1	34	57.6
2	11	18.6
Histology		
Adenocarcinoma	59	100
Gleason score		
6	1	1.7
7	14	23.7
8	15	25.4
9	19	32.2
10	3	5.1
NA	7	11.9
Surgery		
Yes	10	16.9
No	46	78.0
Uknown	3	5.1
Radiotherapy		
Radical Palliative Both Chemo + RT	5 6 3 3	8.5 10.2 5.1 5.1
Total	42	71.2
Overview of previous therapy		
Androgen Receptor Targeted Agents Yes No	59 -	100 -
Chemotherapy Taxane-based Non taxane-based	59 -	100 -
Cabazitaxel treatment		
2nd Line	14	23.7
≥3rd Line	45	76.3
Visceral metastasis		
Yes/Visceral only	14/5	23.7/8.4
No	45	76.2
Baseline PSA (ng/mL)		
<1.0	5	8.5
1.1–4.0	3	5.1
4.1–10.0	4	6.8
10.1–20.0	6	10.2

**Table 2 cancers-15-04511-t002:** CTCs status at baseline, post 1st cycle and at disease progression (PD).

Detectable CTCs—CS ≥ 1 *	Baseline (n = 57) N (%)	Post 1st Cycle (n = 40) N (%)	At PD (n = 20) N (%)
Negative	9 (15.8)	10 (25.0)	3 (15.0)
Positive	48 (84.2)	30 (75.0)	17 (85.0)
Detectable CTCs—CS ≥ 5 *			
Negative	20 (35.1)	16 (40.0)	7 (35.0)
Positive	37 (64.9)	24 (60.0)	13 (65.0)

* per 7.5 mL of peripheral blood.

**Table 3 cancers-15-04511-t003:** OS according to CTC status at baseline.

	CTCs—CS ≥ 1 * (n = 57)			CTCs—CS ≥ 5 * (n = 57)		
	(−) n = 9	(+) n = 48		(−) n = 20	(+) n = 37	
Kaplan–Meier			*p*-Value			*p*-Value
Events	2	29	0.140 (Log-rank test)	5	26	0.002 (Log-rank test)
Median	16.4	12.6		** 28.0	8.8	
Min-Max	3.7–33.9	1.2–30.3		2.5–33.9	1.2–30.3	
95% CI	1.8–31.0	7.2–18.0		11.4–44.6	3.1–14.5	

* per 7.5ml of peripheral blood; ** Estimated (not actual).

**Table 4 cancers-15-04511-t004:** OS according to CTC status after the first chemotherapy cycle.

	CTCs—CS ≥ 1 * (n = 40)			CTCs—CS ≥ 5 * (n = 40)		
	(−) n = 10	(+) n = 30		(−) n = 16	(+) n = 24	
Kaplan–Meier			*p*-Value			*p*-Value
Events	2	20	0.047 (Log-rank test)	4	18	0.005 (Log-rank test)
Median	Not est.	11.3		** 28.0	8.8	
Min-Max	3.7–33.9	1.2–29.4		3.7–33.9	1.2–29.4	
95% CI	-	3.8–18.7		8.7–47.3	2.3–15.3	

* per 7.5ml of peripheral blood; ** Estimated (not actual).

**Table 5 cancers-15-04511-t005:** Phenotypic characterization of CTCs at baseline and after 1 cycle of cabazitaxel.

	Baseline		Post Treatment	
	N of + Patients (%)	Median (Range)	N of + Patients (%)	Median (Range)
CellSearch	37/57 (64.9%)	13 (0–3107)	24/41 (58.5%)	8 (0–1737)
CK^+^/Ki67^+^	30/35 (85.7%)	3 (0–177)	15/21 (71.4%)	3 (0–35)
CK^+^/Ki67^−^	5/35 (14.2%)	5 (0–25)	6/21 (28.5%)	2 (0–25)
CK^+^/M30^+^	7/35 (20.0%)	1 (0–2)	2/21 (9.5%)	1 (0–3)
CK^+^/M30^−^	28/35 (80.0%)	7 (0–28)	18/21 (85.7%)	5 (0–48)
CK^+^/Vim^+^	33/35 (94.3%)	4(0–45)	20/21 (95.2%)	6 (0–13)
CK^+^/Vim^−^	2/35 (5.7%)	2(0–6)	1/21 (4.8%)	1 (0–5)

*p* value: Baseline vs. Post treatment: No statistically significant observation.

**Table 6 cancers-15-04511-t006:** Univariate and multivariate analysis.

	Univariate Analysis				Multivariate Analysis			
	PFS		OS		PFS		OS	
	HR (95% CI)	Sig.	HR (95% CI)	Sig.	HR (95% CI)	Sig.	HR (95% CI)	Sig.
CellSearch at baseline (≥5 vs. <5 CTCs)	-	-	4.0 (1.5–10.5)	0.005	-	-	-	-
CellSearch post 1st cycle (≥5 vs. <5 CTCs)	-	-	4.4 (1.5–12.9)	0.008	-	-	-	-
CK^+^/Ki67^+^ vs. CK^+^/Ki67^−^ at baseline	-	-	-	-	-	-	-	-
CK^+^/Ki67^+^ vs. CK^+^/Ki67^−^ post 1st	-	-	-	-	-	-	-	-
CK^+^/M30^−^ vs. CK^+^/M30^+^ at baseline	9.0 (2–41)	0.005	11.6 (2.3–59.2)	0.003	9.0 (2–41)	0.005	-	-
CK^+^/M30^−^ vs. CK^+^/M30^+^ post 1st	-	-	-	-	-	-	-	-
CK^+^/Vim^+^ vs. CK^+^/Vim^−^ at baseline	-	-	-	-	-	-	-	-
CK^+^/Vim^+^ vs. CK^+^/Vim^−^ post 1st	-	-	-	-	-	-	-	-

## Data Availability

The data generated in this study are available upon reasonable request from the corresponding author.

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
