# Peer review of "Prognostic Role of Circulating Tumor Cells in Patients with Metastatic Castration-Resistant Prostate Cancer Receiving Cabazitaxel: A Prospective Biomarker Study"

_cancers, 2023, doi:10.3390/cancers15184511_

Round 1
Reviewer 1 Report
This manuscript studies whether circulating tumor cells (CTCs) of metastatic castration‐resistant prostate cancer (mCRPC) patients constitute an early predictor of response to cabazitaxel, thus potentially benefiting future mCRPC patients. Yet, some additions/changes as detailed below are recommended.
It would be helpful to briefly discuss potential treatment options after an early CTC‐informed cabazitaxel discontinuation as suggested here, especially considering that according to Table 1, cabazitaxel is the third line (or even later) treatment for more than 75% of the patients tested.
It is suggested to place the tables and figures next to their corresponding description for ease of reading instead of in a separate Section 3.8. Also, the arrows in Table 3 seem unnecessary.
Just curious, instead of analyzing CTC positive or negative patient numbers vs. OS, is it possible to plot absolute CTC numbers against OS or PFS? What would the plot look like? Any trend there?
The following minor changes/clarifications would be really appreciated.
(1) The word “blod” in line 35 seems meant to be “blood.”
(2) Does the phrase “accounting for 11% of all cancer deaths in the US during 2032” in line 69 meant to be “predicted to count for 11% of all cancer deaths in the US during 2032”?
(3) A “)” seems missing in line 291.
(4) A “.” Is missing in lines 387 and 445.
Author Response
Dear reviewer,
We appreciate the time and efforts of the referee in reviewing this manuscript.
The manuscript has been revised on a point-by-point basis addressing reviewer comments. All the changes are also highlighted in green in the revised manuscript. We believe that the revised version can meet the journal publication requirements.
Response to Comments from Reviewer 1
Comments and Suggestions for Authors:
This manuscript studies whether circulating tumor cells (CTCs) of metastatic castration‐resistant prostate cancer (mCRPC) patients constitute an early predictor of response to cabazitaxel, thus potentially benefiting future mCRPC patients. Yet, some additions/changes as detailed below are recommended.
It would be helpful to briefly discuss potential treatment options after an early CTC‐informed cabazitaxel discontinuation as suggested here, especially considering that according to Table 1, cabazitaxel is the third line (or even later) treatment for more than 75% of the patients tested.
It is suggested to place the tables and figures next to their corresponding description for ease of reading instead of in a separate Section 3.8. Also, the arrows in Table 3 seem unnecessary.
Just curious, instead of analyzing CTC positive or negative patient numbers vs. OS, is it possible to plot absolute CTC numbers against OS or PFS? What would the plot look like? Any trend there?
Response 1:
1) “It would be helpful to briefly discuss potential treatment options after an early CTC‐informed cabazitaxel discontinuation..”
Response: Thank you for your input. Treatment options have been added in the discussion section (434-441).
2) “It is suggested to place the tables and figures next to their corresponding description for ease of reading instead of in a separate Section 3.8”
Response: Tables and figures have been placed next to their corresponding description.
3) “Just curious, instead of analyzing CTC positive or negative patient numbers vs. OS, is it possible to plot absolute CTC numbers against OS or PFS”
Response: Thank you for your suggestion. As it is already mentioned in the discussion section, “Danila et al., reported that the number of CTCs at baseline correlated strongly with survival, but no threshold effect could be identified [24], suggesting that CTC counts are important as a continuous variable in predicting the clinical outcome of patients with mCRPC”. However, in the landmark study that led to the FDA clearance of CTC enumeration using the CellSearch platform in patients with CRPC the cutoff values of ≥5 CTC/7.5 mL of PB versus <5 CTC/7.5ml of PB were used to define unfavorable and favorable groups. This cutoff was prospectively evaluated, validated and used in several clinical trials, as indicated in our paper. Indeed, accurate assessment of the actual number of CTCs is of paramount importance when determining response to treatment and prospective trials evaluating CTC enumeration as response and surrogate biomarkers of response in PC are currently ongoing. In line with these studies and to eliminate inter- and intra-operator bias between the different research efforts we used the cutoff values of ≥5 CTC/7.5 mL of PB in order to stratify patients into favorable and not-favorable groups. Moreover, we tested the prognostic value of detecting any number of CTC (cutoff ≥1 CTC/7.5 mL of PB).
We are grateful to the reviewer for their critical contributions and crucial comments. We hope that we have succeeded in addressing all the issues raised by the reviewer and that you find our research suitable for publications in the Cancers.
Comments on the Quality of English Language:
The following minor changes/clarifications would be really appreciated.
- The word “blod” in line 35 seems meant to be “blood.”
Response: The word “blod” has been corrected
- Does the phrase “accounting for 11% of all cancer deaths in the US during 2032” in line 69 meant to be “predicted to count for 11% of all cancer deaths in the US during 2032”?
Response:
- A “)” seems missing in line 291.
Response: parenthesis has been corrected
- A “.” Is missing in lines 387 and 445.
Response: dots have been inserted
Best regards,
Filippos Koinis
Athanasios Kotsakis
Reviewer 2 Report
CRPC has a poor prognosis however the outcome can vary depending on the extent of the disease, the patient's overall health, and the effectiveness of available treatments. Treatment options for castration-resistant prostate cancer may include chemotherapy, immunotherapy, targeted therapies, radionuclide therapy, and palliative care to manage symptoms and improve quality of life. As research continues new markers are needed to select optimal treatment sequence for each patients. This study focuses on CTCs count in CRPC patients treated with Cabazitaxel. The study is well designed and well conducted. The data is well presented. My only complaint is that there was a similiar study about characteriazation of CTCs in CRPC patients treated with Cabazitaxel and authors of this study did not mention it. Please mention it in discussion section. (de Kruijff IE, Sieuwerts AM, Onstenk W, Kraan J, Smid M, Van MN, van der Vlugt-Daane M, Hoop EO, Mathijssen RHJ, Lolkema MP, de Wit R, Hamberg P, Meulenbeld HJ, Beeker A, Creemers GJ, Martens JWM, Sleijfer S. Circulating Tumor Cell Enumeration and Characterization in Metastatic Castration-Resistant Prostate Cancer Patients Treated with Cabazitaxel. Cancers (Basel). 2019 Aug 20;11(8):1212. doi: 10.3390/cancers11081212. PMID: 31434336; PMCID: PMC6721462.)
Please check some minor spelling mistakes.
Author Response
Dear reviewer,
We appreciate the time and efforts in reviewing this manuscript.
The manuscript has been revised on a point-by-point basis addressing reviewer comments. All the changes are also highlighted in green in the revised manuscript. We believe that the revised version can meet the journal publication requirements.
Response to Comments from Reviewer 1
Comments and Suggestions for Authors:
CRPC has a poor prognosis however the outcome can vary depending on the extent of the disease, the patient's overall health, and the effectiveness of available treatments. Treatment options for castration-resistant prostate cancer may include chemotherapy, immunotherapy, targeted therapies, radionuclide therapy, and palliative care to manage symptoms and improve quality of life. As research continues new markers are needed to select optimal treatment sequence for each patients. This study focuses on CTCs count in CRPC patients treated with Cabazitaxel. The study is well designed and well conducted. The data is well presented. My only complaint is that there was a similiar study about characteriazation of CTCs in CRPC patients treated with Cabazitaxel and authors of this study did not mention it. Please mention it in discussion section. (de Kruijff IE, Sieuwerts AM, Onstenk W, Kraan J, Smid M, Van MN, van der Vlugt-Daane M, Hoop EO, Mathijssen RHJ, Lolkema MP, de Wit R, Hamberg P, Meulenbeld HJ, Beeker A, Creemers GJ, Martens JWM, Sleijfer S. Circulating Tumor Cell Enumeration and Characterization in Metastatic Castration-Resistant Prostate Cancer Patients Treated with Cabazitaxel. Cancers (Basel). 2019 Aug 20;11(8):1212. doi: 10.3390/cancers11081212. PMID: 31434336; PMCID: PMC6721462.)
Response 1:
Thank you for your advice. We had mentioned that “our cutoff analysis did confirm the previously reported negative prognostic role of more than 5 CTCs/7.5 mL of PB for patients with mCRPC treated with cabazitaxel.”, but by mistake we had not added the citation. We have made an addition briefly discussing the results from this trial and of course added the citation.
We are grateful to the reviewer for their critical contributions and crucial comments. We hope that we have succeeded in addressing all the issues raised by the reviewer and that you find our research suitable for publications in the Cancers.
Comments on the Quality of English Language:
- Please check some minor spelling mistakes.
Response:
A native speaker review our manuscript and made the appropriate corrections.
Best regards,
Filippos Koinis
Athanasios Kotsakis
Round 2
Reviewer 1 Report
Many thanks for the authors’ patient explanation and amendments. Really appreciate them. Yet, I do have some further concerns and would really appreciate the authors’ input.
Additional examples of treatment options available to patients after an early CTC-informed cabazitaxel discontinuation are suggested, for example, sipuleucel-T, PARP inhibitors, and Radium-223, if appropriate. Some reasons here are (1) 177Lu-PSMA-617 has its own biomarker anyway (PSMA positive); (2) microsatellite instability-high (MSI-H) is considered as a very small proportion of prostate cancer patients (see, PMID: 36199275); (3) metronomic cyclophosphamide (mCyc) seems still in the validation/development stage and its use “in mCRPC has fallen out of favor with development of novel therapies” (see, Journal of Clinical Oncology 41, no. 16_suppl (June 01, 2023) e17041-e17041); and (4) effectiveness of a platinum-based therapy in treating mCRPC patients with DNA repair gene aberrations is still under evaluation (see, PMID: 33112397).
In addition, de Kruijff et al as cited by the authors seems have studied CTCs at baseline and after 6 weeks of cabazitaxel therapy. See more details in Section 2.3 of de Kruijff et al. Kindly highlight the advances made in the current manuscript over those disclosed in de Kruijff et al.
As indicated previously, does the phrase “accounting for 11% of all cancer deaths in the US during 2032” in line 69 meant to be “predicted to count for 11% of all cancer deaths in the US during 2032”?
The word “have” in line 145 seems appropriate after "must" and does not require correction.
Author Response
We appreciate the time and efforts of the referee in reviewing this manuscript.
The manuscript has been revised on a point-by-point basis addressing reviewer comments. All the changes are also highlighted in green in the revised manuscript. We believe that the revised version can meet the journal publication requirements.
Response to Comments from Reviewer 1
Comments and Suggestions for Authors:
Comments and Suggestions for Authors
Many thanks for the authors’ patient explanation and amendments. Really appreciate them. Yet, I do have some further concerns and would really appreciate the authors’ input.
Additional examples of treatment options available to patients after an early CTC-informed cabazitaxel discontinuation are suggested, for example, sipuleucel-T, PARP inhibitors, and Radium-223, if appropriate. Some reasons here are (1) 177Lu-PSMA-617 has its own biomarker anyway (PSMA positive); (2) microsatellite instability-high (MSI-H) is considered as a very small proportion of prostate cancer patients (see, PMID: 36199275); (3) metronomic cyclophosphamide (mCyc) seems still in the validation/development stage and its use “in mCRPC has fallen out of favor with development of novel therapies” (see, Journal of Clinical Oncology 41, no. 16_suppl (June 01, 2023) e17041-e17041); and (4) effectiveness of a platinum-based therapy in treating mCRPC patients with DNA repair gene aberrations is still under evaluation (see, PMID: 33112397).
In addition, de Kruijff et al as cited by the authors seems have studied CTCs at baseline and after 6 weeks of cabazitaxel therapy. See more details in Section 2.3 of de Kruijff et al. Kindly highlight the advances made in the current manuscript over those disclosed in de Kruijff et al.
As indicated previously, does the phrase “accounting for 11% of all cancer deaths in the US during 2032” in line 69 meant to be “predicted to count for 11% of all cancer deaths in the US during 2032”?
Response 1:
- “Additional examples of treatment options available to patients after an early CTC-informed cabazitaxel discontinuation are suggested, for example, sipuleucel-T, PARP inhibitors, and Radium-223, if appropriate. Some reasons here are (1) 177Lu-PSMA-617 has its own biomarker anyway (PSMA positive); (2) microsatellite instability-high (MSI-H) is considered as a very small proportion of prostate cancer patients (see, PMID: 36199275); (3) metronomic cyclophosphamide (mCyc) seems still in the validation/development stage and its use “in mCRPC has fallen out of favor with development of novel therapies” (see, Journal of Clinical Oncology 41, no. 16_suppl (June 01, 2023) e17041-e17041); and (4) effectiveness of a platinum-based therapy in treating mCRPC patients with DNA repair gene aberrations is still under evaluation (see, PMID: 33112397).”
Response: Thank you for your input. Treatment options have been added in the discussion section (434-441). These are still options in treating patients that have already received at least three prior lines of treatment (see NCCN guidelines: https://www.nccn.org/professionals/physician_gls/pdf/prostate.pdf). As far as cyclophosphamide is concerned, we think that it is still an option especially in patients that are not included in the previous described molecular-defined subgroups. Major and long lasting responses have been described (https://jnccn.org/pdfviewer/full/journals/jnccn/11/8/article-p911.xml?bg=GREY_COLOR_LIGHT_VARIANT&mgs=PRIMARY_COLOR&mge=PRIMART_COLOR_DARK_VARIANT) and an immunomodulatory effect of cyclophosphamide in pts with mCRPC has been suggested (https://www.annalsofoncology.org/article/S0923-7534(19)56679-0/fulltext). In the latter report, a clinical benefit in 75% of those pts has been observed. This is important given the lack of treatment options in this setting (PMID: 36672410). Our group has also seen clinically significant responses especially in patients with AVPC. Another option that is of paramount importance is, as we have stated, enrollment in a clinical trial. In any case, we have also stated that “Whether such a strategy could lead to an OS benefit remains to be answered by large randomized prospective clinical trials.” Moreover, discontinuing a drug that has toxicity and is not beneficial for the patient is also important (“…..and by minimizing the toxicity from ineffective treatment”).
- “In addition, de Kruijff et al as cited by the authors seems have studied CTCs at baseline and after 6 weeks of cabazitaxel therapy. See more details in Section 2.3 of de Kruijff et al. Kindly highlight the advances made in the current manuscript over those disclosed in de Kruijff et al.”
Response: Thank you for your comment. We have used and tested the prognostic significance of different cutoffs for defining CTC-positive and CTC-negative patient populations. Indeed, our cutoff analysis confirmed the previously reported negative prognostic role of more than 5 CTCs/7.5 mL of PB (and not 1 CTCs/7.5 ml of PB) for patients with mCRPC treated with cabazitaxel. Moreover, the second blood sample was obtained 21 days after the administration of the first chemotherapy cycle and not after the second chemotherapy cycle as de Kruijff et al., did. The notion behind this was to evaluate the prognostic significance of CTC counts and CTC conversion at an earlier timepoint (after 1 and not after 2 chemotherapy cycles). Interestingly, a positive CTC status after the administration of one chemotherapy cycle strongly correlated with poorer OS irrespectively of the used cut-off (p=0.047, p=0.005). As we have stated “it was yet to be determined whether CTCs’ detection after 1 cycle of treatment constitutes an early predictor of response to cabazitaxel”. Moreover, in addition to CTC enumeration, a deeper molecular interrogation was performed to further appreciate the phenotypic CTC heterogeneity and the effect of cabazitaxel in different CTC subgroups. The detection of non-apoptotic CK+/M30- cells was associated with the clinical outcome, both in terms of PFS and OS. This is the first study demonstrating the prognostic significance of this CTC subgroup in patients with mCRPC.
- “As indicated previously, does the phrase “accounting for 11% of all cancer deaths in the US during 2032” in line 69 meant to be “predicted to count for 11% of all cancer deaths in the US during 2032”?”
Response: Thank you for your comment. We have re-phrased the sentence.
Comments on the Quality of English Language
The word “have” in line 145 seems appropriate after "must" and does not require correction.
Response: The word “have” has been corrected.
We are grateful to the reviewer for the critical contributions and crucial comments. We hope that we have succeeded in addressing all the issues raised by the reviewer and that you find our research suitable for publications in the Cancers.
Best regards,
Filippos Koinis
Athanasios Kotsakis